# Investigation of the Structural Perfection of a LiNbO$_3$:Gd$^{3+}$(0.003):Mg$^{2+}$(0.65 wt.%) Double-Doped Single Crystal Using the Raman Spectra Excited by Laser Lines in the Visible (532 nm) and Near-IR (785 nm) Regions

Nikolay Sidorov [1], Mikhail Palatnikov [1], Alexander Pyatyshev [2,*] and Alexander Skrabatun [2,3]

1   Tananaev Institute of Chemistry—Subdivision of the Federal Research Centre "Kola Science Centre of the Russian Academy of Sciences", "Academic Town", 26a, Murmansk Region, 184209 Apatity, Russia
2   P.N. Lebedev Physical Institute of the Russian Academy of Sciences, Leninskiy Prospekt 53, 119991 Moscow, Russia
3   Physics Department, Faculty of Fundamental Sciences, Bauman Moscow State Technical University, 2nd Baumanskaya St. 5/1, 105005 Moscow, Russia
*   Correspondence: jb_valensia@mail.ru

**Abstract:** A compositionally homogeneous nonlinear optical single crystal of double-doped LiNbO$_3$:Gd$^{3+}$(0.003):Mg$^{2+}$(0.65 wt.%) was obtained. Fine features of the LiNbO$_3$:Gd$^{3+}$(0.003):Mg$^{2+}$(0.65 wt.%) crystal structure were studied from the Raman spectra of the first and second orders upon excitation by laser lines in the visible (532 nm) and near-IR (785 nm) regions. When the Raman spectrum was excited by a 785 nm laser line in the frequency range of 1000–2000 cm$^{-1}$ for the first time, a number of low-intensity lines in the range of 900–2000 cm$^{-1}$, corresponding to the second-order Raman spectrum, were discovered. The same lines also appear in the spectrum upon excitation by a laser line with a wavelength of 532 nm, but their intensities are significantly (by an order of magnitude or more) lower. It is shown that in the structure of the double-doped LiNbO$_3$:Gd$^{3+}$(0.003):Mg$^{2+}$(0.65 wt.%), the crystal oxygen-octahedral clusters MeO$_6$ (Me–Li, Nb, Gd, Mg) are slightly distorted, and in addition, the value R = [Li]/[Nb] ≈ 1 is close to that for a nominally pure stoichiometric crystal.

**Keywords:** lithium niobate single crystal; double doping; structural perfection; Raman scattering; second-order spectrum

## 1. Introduction

An urgent task of modern physical materials science is to obtain highly advanced optical materials with a low photorefraction effect (optical damage) based on a nonlinear optical single crystal of lithium niobate (LiNbO$_3$). Lithium niobate is an oxygen-octahedral phase of variable composition with a wide homogeneity region on the phase diagram [1,2]. The set of optical properties of this material—the magnitudes of electro-optical and nonlinear optical coefficients, the sensitivity to the holographic recording of information, and the possibility of obtaining laser generation with frequency self-doubling—make it universal for optical applications, such as quartz in acoustics. One of the reasons for the universality of lithium niobate as a phase of variable composition is the possibility of controlling its properties over a wide range by varying the composition through doping and changing the stoichiometry, which is especially attractive for the development of integrated optical devices.

Among the properties that strongly depend on the composition is the effect of a photoinduced change in the refractive indices (optical damage), which leads to the distortion of the wave front of a laser beam passing through the crystal [1]. The existence of the photorefraction effect causes two important alternative practical problems: finding ways to suppress its (i.e., obtaining non-photorefractive, optical-damage resistant") compositions

for traditional applications, and the optimization of photorefractive properties; for example, increasing the sensitivity and speed of information recording. For practice, the first task is much more important, since the use of a lithium niobate crystal in nonlinear optics as an active nonlinear laser medium, as well as for converting and modulating laser radiation, remains dominant. This task is especially relevant in the current intensive development of the method of optical frequency conversion in the regime of phase quasi-phase matching on regular domain structures (PPLN—"periodically-poled LiNbO$_3$") [3].

LiNbO$_3$:Mg crystals are currently the most promising for the creation of highly advanced materials for radiation conversion [3–8]. In the case of double doping, when one of the alloying elements is magnesium, it can be used to create optical materials of increased compositional homogeneity with a minimum photorefractive response time and a high resistance to optical damage [9–16]. In highly perfect double-doped LiNbO$_3$ crystals, the fundamental absorption edge shifts to shorter wavelengths, and a noticeable increase in nonlinear optical coefficients is observed vs. one-time doped crystals [11,12,14,16,17]. Co-doping simultaneously with two "non-photorefractive" cations (Mg$^{2+}$ and Gd$^{3+}$) makes it possible to control the ordering of structural units of the cationic sublattice along the polar axis and the polarizability of MeO$_6$ clusters (Me–Li, Nb, vacancy, and impurity metal) more finely than single doping. It also allows one to control the type and concentration of point and complex defects with localized electrons, which determine the magnitude of the photorefraction effect (optical damage) [1,11]. This is due to the fact that both the Mg$^{2+}$ and Gd$^{3+}$ cations occupy mainly lithium positions in LiNbO$_3$ crystals [2]. Therefore, in co-doped LiNbO$_3$:Gd:Mg crystals, there is competition for the lithium positions between the Mg$^{2+}$ and Gd$^{3+}$ cations, which leads to a change in the number and type of defects, which are shallow and deep electron traps that increase the photorefraction effect. There is no such competition in LiNbO$_3$:Mg and LiNbO$_3$:Gd [1,2].

Thus, it is of considerable interest to establish the possibility of controlling the optical and electrical properties of a LiNbO$_3$ crystal by doping LiNbO$_3$ crystals with various cations or groups of cations.

Raman spectroscopy (RS) is a well-known method for studying various organic and inorganic compounds. This method has a high sensitivity, which makes it possible to study objects under difficult conditions. In particular, art objects of various ages [18–20], food [21,22], viruses and bacteria [23–25], recycled polyethylene terephthalate [26], and so on, have previously been studied. The high sensitivity of RS to the slightest changes in the interaction between structural units is an important factor for studying the crystalline perfection of LiNbO$_3$. An important feature of the polar ferroelectric LiNbO$_3$ is that the transverse (TO) and longitudinal (LO) vibrations of the crystal lattice are characterized by a strong interaction with electromagnetic radiation, which excites the Raman spectra [2,27]. As mentioned above, lithium niobate is a photorefractive crystal. In this regard, an additional volumetrically ordered sublattice of nano- and microstructures appears in the illuminated region of the crystal. In this region, there are changes in the refractive index, permittivity, conductivity, and other parameters due to the photoreaction effect. As a result, the Raman scattering lines are shifted in frequency, their intensity changes, and they broaden. For this reason, the Raman spectra obtained upon excitation by lasers with different wavelengths can noticeably differ from each other [28]. This may be due to the different sensitivities of micro- and macrostructures, MeO$_6$ clusters, point defects in the form of irregularly located main and impurity atoms, and the effects of structural disorder and anharmonicity that exist in the LiNbO$_3$ crystal as a photorefractive phase of variable composition [1,2,27], and the effects of radiation over a wide spectral range.

In this work, in the frequency range of 50–4000 cm$^{-1}$, with the excitation of the spectra in the visible (532 nm) and near-IR (785 nm) regions, the full Raman spectrum of a double-doped LiNbO$_3$:Gd$^{3+}$(0.003):Mg$^{2+}$(0.65 wt.%) is measured. Gd$^{3+}$ and Mg$^{2+}$ cations, which have different valences and ionic radii, are the non-photorefractive additives that differently affect the state of the crystal defect sublattice, spontaneous polarization along the polar axis, the geometry of NbO$_6$ oxygen-octahedral clusters, and the effect of photorefraction

(optical damage). Previously, the Raman spectra of LiNbO$_3$:Gd:Mg crystals of different compositions were studied in the literature only in the range of 50–1000 cm$^{-1}$ [27,29]. According to the first-order Raman spectra in the frequency range of 50–1000 cm$^{-1}$ in single-doped LiNbO$_3$:Gd$^{3+}$ and LiNbO$_3$:Mg$^{2+}$ crystals at low concentrations of Gd$^{3+}$ (0.002 wt.%) and Mg$^{2+}$ (0.030–0.078 wt.%), there is an increase in the ordering of structural units of the cationic sublattice along the polar axis, and a noticeable decrease in the photorefraction effect [25,27]. The same effects are observed for heavily doped Mg$^{2+}$ (0.40–0.65 wt.%) crystals of double-doped LiNbO$_3$:Gd$^{3+}$:Mg$^{2+}$ at low concentrations of the second doping component Gd$^{3+}$ (0.001–0.23 wt.%) [27,29]. At high levels of magnesium doping (>3.0 wt.%), additional lines appear in the Raman spectra that conform to pseudoscalar fundamental lattice vibrations of the A$_2$ symmetry type, which are not allowed in the Raman spectra for the R3c symmetry space group [30]. It was also found that a magnification of the Mg amount in single and co-doped LiNbO$_3$ leads to an increase in the Raman intensity on vibrations of the E modes without any spectral shift [9,31,32].

## 2. Materials and Methods

A single crystal of LiNbO$_3$:Gd$^{3+}$(0.003):Mg$^{2+}$(0.65 wt.%) was grown in air from a melt using the Czochralski method on a Kristall 2 (ZavodKristall Ltd., Likino village, Russia) induction growth setup with automatic control of the crystal diameter (Figure 1) in a platinum crucible 75 mm in diameter with a small (2.5 deg/cm) axial gradient in the direction of the polar axis (z-cut, (001)). The single-crystal boule rotation speed was ~16–18 rpm, and the crystal pulling speed was ~0.7 mm/h. In this case, the crystal growth rate was ~1.02–1.04 mm/h. The technological parameters of growth corresponded to the condition of a flat crystallization front. The direct alloying of the congruent melt ([Li]/[Nb] = 0.946) with MgO and Gd$_2$O$_3$ was carried out. To relieve thermoelastic stresses, the grown crystal was subjected to heat treatment in an air atmosphere at 1500 K for 15 h in a high-temperature furnace, Lantan (Voroshilovgradsky zavod electronnogo mashinostroeniya, Voroshilovgrad, USSR).

Growing optically and compositionally uniform doubly doped lithium niobate crystals is a non-trivial technological problem. Doping additives, as a rule, have impurity distribution coefficients that differ (in our case, strongly differ) in magnitude: K$_D$(Mg) ≈ 1.1, K$_D$(Gd) ≈ 0.25 [33]. Consequently, the melt composition near the solidification front during crystal growth can be simultaneously enriched in one alloying component and depleted in another. Thus, the composition of a doped crystal during growth can change significantly from the cone to the end of the crystal, which usually leads to a decrease in its compositional and optical homogeneity. In this case, the characteristics of the crystal can significantly vary within its different parts. To minimize such effects, it is necessary to apply changes in parameters that are natural for the process of crystal growth. These are the speed of rotation and movement of the crystal, temperature gradients in the melt and growth zone, and different conjunctions of these parameters. In our case, such technological methods, including the use of a special design of the thermal unit, which creates small temperature gradients at the crystallization front, the use of low crystallization rates, particular melt preparations before crystal growth, long postgrowth annealing, and suitable conditions for the electrothermal treatment of the crystal, are effective.

When growing a single crystal of double-doped LiNbO$_3$:Gd$^{3+}$(0.003):Mg$^{2+}$(0.65 wt.%), we used the design of a thermal unit with double "warming", which allows for the creation of an isothermal zone in the volume of the platinum screen for the postgrowth annealing of the crystal, and the growth of the crystal under the conditions of a small temperature gradient at the crystallization front. The design of the thermal unit was similar to that used in [34]. Figure 1b shows the grown LiNbO$_3$:Gd$^{3+}$(0.003):Mg$^{2+}$(0.65 wt.%) crystal.

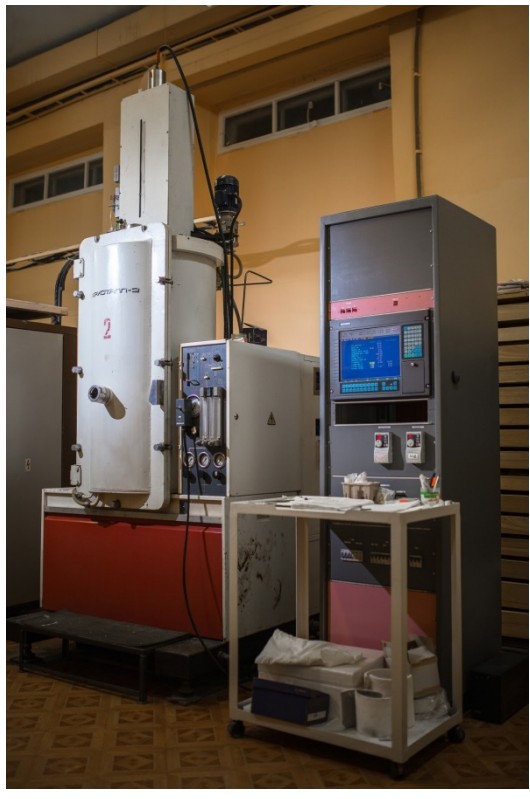

(**a**)

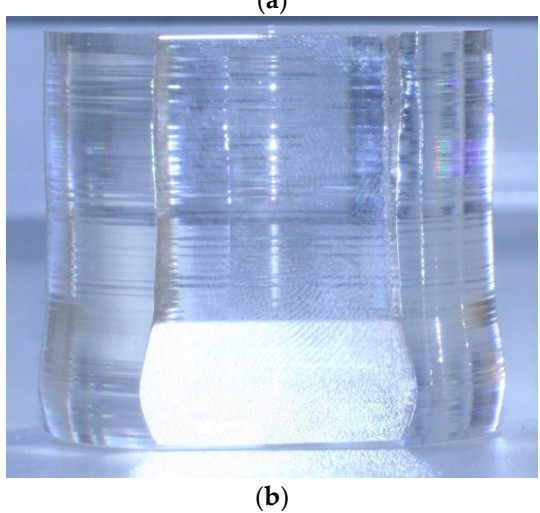

(**b**)

**Figure 1.** Kristall 2 growth setup for growing lithium niobate crystals using the Czochralski method—(**a**); crystal LiNbO$_3$:Gd$^{3+}$(0.003):Mg$^{2+}$ (0.65 wt.%) (**b**). Figure is reproduced with the permission of Elsevier from the paper [34].

The crystal growth process was finished when the LiNbO$_3$:Gd:Mg crystal's mass of less than 275 g was reached. In this case, about ~25% of the total mass of the melt crystallized. The parameters of the LiNbO$_3$:Gd:Mg crystal growth process (pulling speed, rod rotation speed, and temperature gradient at the crystallization front) were selected experimentally, based on the need to obtain a flat crystallization front, which should ensure a sufficiently high structural perfection of the crystal. The estimated mass during the growth of the LiNbO$_3$:Gd$^{3+}$(0.003):Mg$^{2+}$(0.65 wt.%) crystal was determined from the readings of the weight sensor of the growth setup; the accurate mass of the crystal was determined by weighing the grown crystal after separating the seed from the crystal boule. The grown crystals had a flat crystallization front and geometric dimensions: diameter (Ø) ≈ 38 mm,

length of the cylindrical part $L_c \approx 40$ mm. Impurities were introduced into the charge in the form of high-purity oxides of MgO and $Gd_2O_3$ with a concentration of foreign admixtures of no greater than $3 \times 10^{-4}$ wt.%, followed by thorough mixing. The melt before the start of crystal growth was kept for 8–11 h under superheating conditions by 180–200 °C relative to the melting temperature ($T_{melt}$ = 1263 °C) of lithium niobate, to homogenize the impurities in the melt. After growth, the Mg crystal was annealed at 1200 °C in a growth setup for 10 h, and then cooled at a rate of ~50 deg/h. Long-term post-growth annealing in the isothermal zone (same as in [34]) under a platinum screen is required to homogenize the composition of the doped crystal, and to remove the thermal and mechanical stresses.

The monodomainization of the $LiNbO_3$:Gd:Mg crystal was carried out via high-temperature electrodiffusion annealing, namely by applying a constant electric voltage to the polar cuts of the crystal during its cooling, at a rate of 20 deg/h in the temperature range ~1230–870 °C.

For the synthesis of the lithium niobate charge, $Nb_2O_5$ grade A was used, and produced using Technical Specifications No. 1763-025-00545484-2000 at Solikamsk magnezium works (Solikamsk, Russia), and with $Li_2CO_3$ of high purity, with an amount of foreign admixtures of no greater than $3 \times 10^{-4}$ wt.%. From these initial components, a granular charge of a congruent composition ($[Li_2O]$ = 48.6 mol.%) with a high bulk density (~3.4 g/cm$^3$) was obtained, using the synthesis–granulation method. The preparation of the charge is described in detail in [35].

To estimate the impurity content in the $LiNbO_3$:$Gd^{3+}$(0.003):$Mg^{2+}$(0.65 wt.%) crystal, and the distribution features of the change in their concentration along the length of the crystal, samples were studied from the upper (cone) and lower cylindrical parts of the boule. Inductively coupled plasma atomic emission spectroscopy (ICP-AES) on an Optima 8300 ICP-OES (Mg) (PerkinElmer, Waltham, MA, USA) spectrometer, and atomic absorption spectrometry (AAS) on a Kvant-FA instrument (Zn) (GRANAT, Saint Petersburg, Russia) were used to determine the concentration of dopants in milled samples. Table 1 shows the admixture concentrations of the granular mixture and the studied $LiNbO_3$:$Gd^{3+}$(0.003):$Mg^{2+}$(0.65 wt.%) single crystal.

**Table 1.** Concentrations of admixtures (C, wt.%) in the granular charge, as well as in the cone and end parts of the $LiNbO_3$:$Gd^{3+}$(0.003):$Mg^{2+}$(0.65 wt.%) crystal.

| Admixture | Concentration $C \cdot 10^{-3}$, wt.% | | |
|---|---|---|---|
| | In Charge | Cone of the Crystal | End of the Crystal |
| Mn | <0.2 | <0.2 | <0.2 |
| Ni | <0.3 | <0.3 | <0.3 |
| Al | <0.3 | <0.3 | 1 |
| Fe | <0.3 | 0.32 | 0.38 |
| Cr, Cu, V | 0.3 | 0.3 | 0.3 |
| Pb, Sn | <0.5 | <0.5 | <0.5 |
| Bi | 0.5 | 0.5 | 0.5 |
| Mg | 0.5 | 0.53 | 0.58 |
| Si, Ti, Mo, Ca, Co | 1 | 1 | 1 |
| Sb | 2.1 | 1.7 | 2 |
| Zr | <10 | 10 | 10 |

The sample for studying the Raman spectra was a parallelepiped with dimensions of $4.93 \times 5.98 \times 8.41$ mm$^3$, and edges coinciding with the direction of the principal crystallographic axes. The faces of the parallelepiped were thoroughly burnished.

To record the Raman spectra in the visible region, we used a BWS465-532S i-Raman Plus spectrometer (B&W Tek, Plainsboro Township, NJ, USA), which has a wavelength of 532 nm and allows for the recording of spectra in the range of 50–4000 cm$^{-1}$. The power of laser radiation during the registration of spectra was 30 mW. The numerical aperture was ≈0.22. To record the Raman spectra in the near-IR region, a BWS465-785H



i-Raman Plus spectrometer (B&W Tek, Plainsboro Township, NJ, USA) was used, with an excitation wavelength of 785 nm and with the allowance of recording spectra in the range of 50–2850 cm$^{-1}$. The power of laser radiation during the registration of spectra was 340 mW. The numerical aperture was ≈0.22. The laser spot size at the focus was 85 μm. All spectra were recorded at room temperature using backscattering geometry. In order to minimize the local influence of the exciting laser radiation, in each experiment, we selected the optimal regimes of radiation, focusing on the crystals under study and the accumulation time of the useful signal.

## 3. Results and Discussion

The composition of the $LiNbO_3:Gd^{3+}(0.003):Mg^{2+}(0.65$ wt.%) crystal on the phase diagram of the $Nb_2O_5$-$Li_2O$ system is within the homogeneity region (solid solution region). Therefore, the unit cell of the $LiNbO_3:Gd^{3+}(0.003):Mg^{2+}(0.65$ wt.%) crystal, such as the unit cell of the ferroelectric phase of the nominally pure $LiNbO_3$ crystal, is characterized by the space symmetry group $C_{3V}^6$ (R3c) and contains two formula units (10 atoms) [2]. In this case, the doping of the $Gd^{3+}$ and $Mg^{2+}$ cations in the structure of the $LiNbO_3$ crystal violate the order of alternation of the $Li^+$ and $Nb^{5+}$ cations, and vacancies (V) along the polar axis, and it distorts the geometry of the oxygen-octahedral $MeO_6$ clusters, which is characteristic of a nominally pure crystal of congruent composition. The main $Li^+$ and $Nb^{5+}$ ions, as well as the doping of the $Mg^{2+}$ and $Gd^{3+}$ ions, occupy the $C_3$ position, while the $O^{2-}$ ions occupy the $C_1$ position. The phonon dispersion curve thus has 30 vibrational branches, of which 27 are optical and 3 are acoustic. The optical vibrational representation of a $LiNbO_3$ crystal has the following form:

$$\Gamma = 5A_1(z) + 5A_2 + 10E(x, y).$$

In Raman scattering and IR absorption at **k** = 0 (at the center of the Brillouin zone), active $4A_1(z) + 9E(x, y)$ dipole-active fundamental vibrations occur, respectively, along and perpendicular to the polar axis. Due to the polar nature of all optical vibrations in a crystal, they are split into longitudinal (LO) and transverse (TO). Thus, in the Raman spectra, under the condition of the propagation of phonons along the main crystallographic axes, taking into account the LO-TO splitting, 26 lines corresponding to fundamental phonons should appear [2]. There are also $A_1(z) + E(x, y)$ acoustic and $5A_2$ optically inactive fundamental vibrations that should not be detected in the Raman and IR absorption spectra. It follows from the form of the Raman tensors [16] that only nondegenerate phonons of $A_1(z)$ symmetry appear in the polarization (*zz*); in polarizations (*xy*), (*xz*), (*yx*), (*yz*), (*zx*), (*zy*)—only doubly degenerate phonons of $E(x, y)$—symmetry types. In polarizations (*xx*) and (*yy*), phonons $A_1(z)$ and $E(x, y)$ of symmetry types must be present simultaneously.

Figures 2 and 3 show the Raman spectra of the $LiNbO_3:Gd^{3+}(0.003):Mg^{2+}(0.65$ wt.%) crystal in some backscattering geometries, recorded upon excitation by laser lines with a wavelength of 532 and 785 nm, respectively. The experimentally observed frequencies and their assignments are presented in Table 2. The assignment of experimentally recorded frequencies was carried out on the basis of widely cited articles [36,37], in which crystals of various stoichiometry, powders, and solid solutions were studied using RS and IR spectroscopy at various temperatures. For some registered Raman lines, there is an alternative assignment, shown in parentheses. As this table shows, we did not assign all registered Raman lines. Thus, upon the excitation of Raman radiation via laser radiation in the near-IR range (785 nm), a line with a frequency of 615 cm$^{-1}$ is observed. This line is close to the calculated value (610 cm$^{-1}$) for the $4A_1(z)TO$ mode [37]. In one of the first works on Raman spectroscopy in lithium niobate [38], a line with a frequency of 603 cm$^{-1}$ was observed, which the authors attributed to an $A_1(z)TO + E(x, y)TO$ vibration. Other theoretical calculations [39–41] refer to the 603 cm$^{-1}$ line for the fully symmetrical fundamental $A_1(z)TO$-mode. The difference in the frequency of this line can be attributed to the choice of the calculation model, as well as the presence in our work of a lithium niobate crystal co-doped with magnesium and gadolinium. When excited by laser radiation in the

visible range (532 nm), a line with a frequency of 732 cm$^{-1}$ was registered in the Raman spectrum. Taking into account the double doping of the studied crystal and the error of the used spectrometer, it can be attributed to second-order Raman scattering [42].

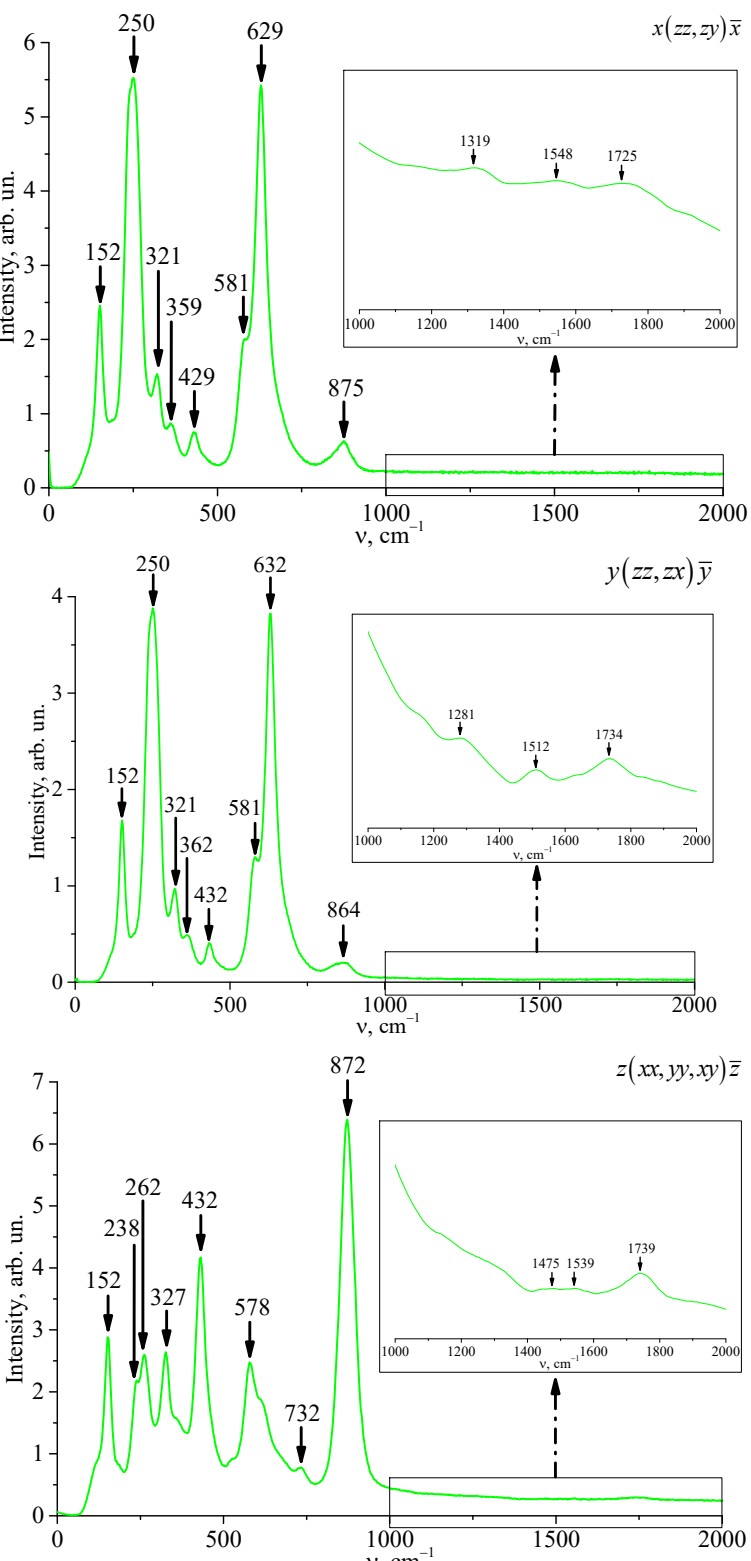

**Figure 2.** Raman spectra in the band of 50–2000 cm$^{-1}$ for the backscattering geometry of a LiNbO$_3$:Gd$^{3+}$(0.003):Mg$^{2+}$(0.65 wt.%) crystal obtained upon excitation by a laser line with a wavelength of 532 nm.

**Table 2.** The frequencies of transverse (TO) and longitudinal (LO) fundamental polar vibrations observed in the Raman spectrum of the $LiNbO_3:Gd^{3+}(0.003):Mg^{2+}(0.65$ wt.%) crystal obtained in this work, as well as their assignment using [36,37]. Below are the line frequencies corresponding to the second-order spectrum.

| Assignment | $\lambda_0 = 532$ nm | | | $\lambda_0 = 785$ nm | | |
|---|---|---|---|---|---|---|
| | $\nu$, cm$^{-1}$ | | | | | |
| | $x(zz,zy)\bar{x}$ | $y(zz,zx)\bar{y}$ | $z(xx,yy,xy)\bar{z}$ | $x(zz,zy)\bar{x}$ | $y(zz,zx)\bar{y}$ | $z(xx,yy,xy)\bar{z}$ |
| 1E(TO) | 152 | 152 | 152 | 153 | 153 | 153 |
| 1E(LO) | | | | 186 | 182 | 188 |
| 2E(LO) | | | 238 | 237 | 237 | 237 |
| 1A$_1$(TO) | 250 | 250 | | 253 | 256 | |
| 3E(TO) | | | 262 | | | 262 |
| 3E(LO) | | | | | | 296 |
| 4E(TO) | 321 | 321 | 327 | 322 | 322 | 328 |
| 5E(TO) (5E(LO)) | 359 | 362 | | 367 | 365 | 363 |
| 6E(LO) (7E(TO)) | 429 | 432 | 432 | 436 | 433 | 429 |
| 8E(TO) | 581 | 581 | 578 | 578 | | |
| 4A$_1$(TO) | 629 | 632 | | 629 | 629 | |
| 4A$_1$(LO) (9E(LO)) | 875 | 864 | 872 | 876 | 876 | 875 |
| | | | | 1046 | 1036 | 1041 |
| | | | | 1122 | | 1131 |
| | | | | 1202 | | |
| | | 1281 | | 1291 | 1279 | 1291 |
| | 1319 | | | 1351 | 1398 | |
| | | | | 1411 | | |
| | | | 1475 | | | |
| | 1548 | 1512 | 1539 | 1523 | 1525 | 1522 |
| | | | | 1595 | | |
| | 1725 | 1734 | 1739 | 1709 | 1720 | 1720 |
| | | | | 1853 | | 1862 |
| | | | | 1963 | | |

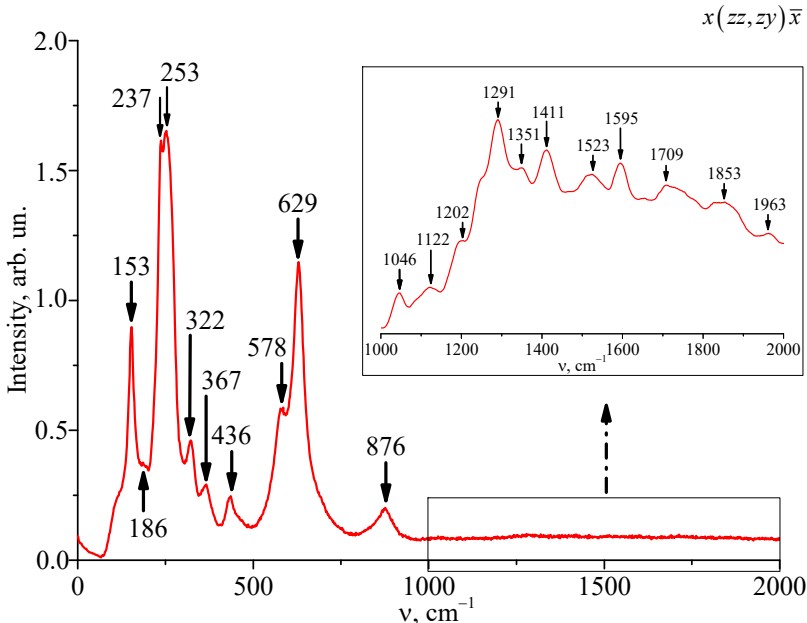

**Figure 3.** *Cont.*

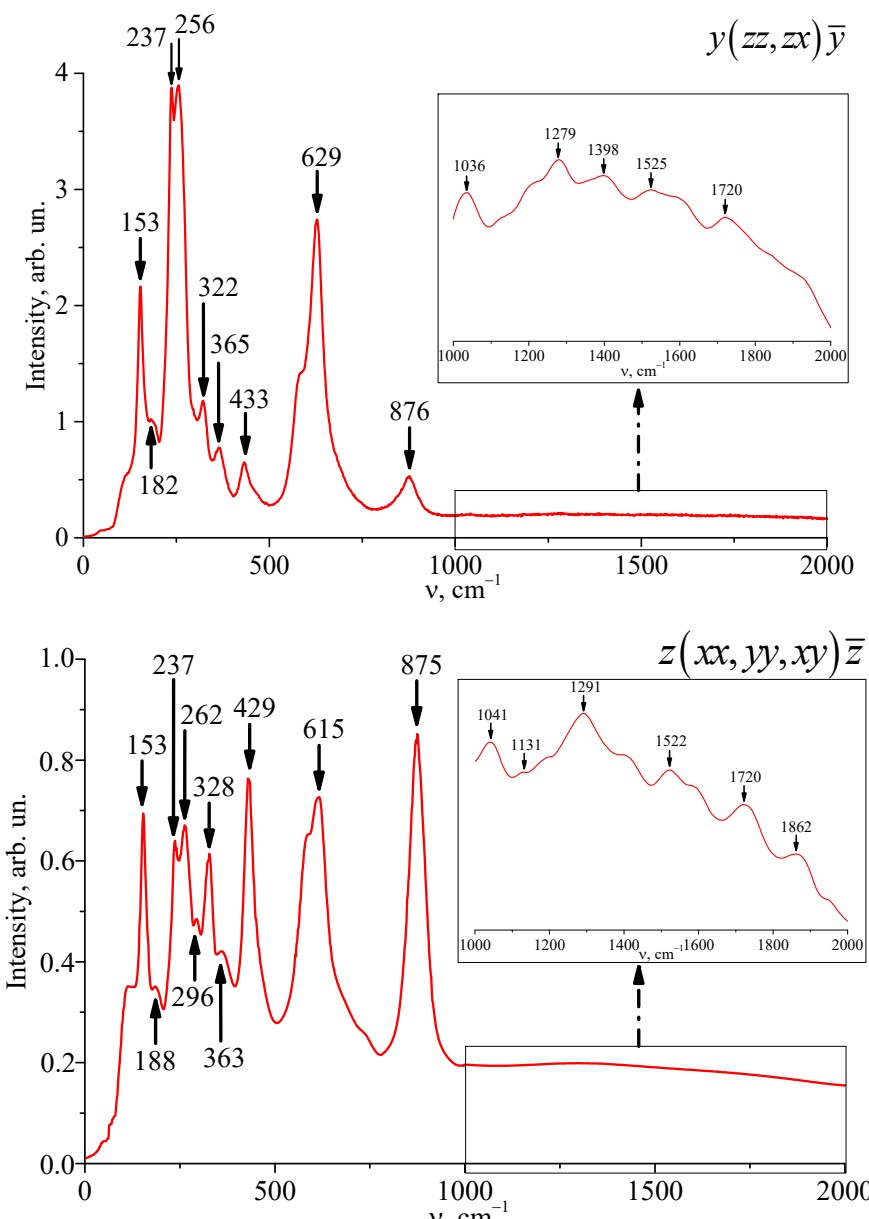

**Figure 3.** Raman spectra in the band of 50–2000 cm$^{-1}$ for the backscattering geometry of the LiNbO$_3$:Gd$^{3+}$(0.003):Mg$^{2+}$(0.65 wt.%) crystal obtained upon excitation by a laser line with a wavelength of 785 nm.

According to Figure 2, the registered Raman spectra contain intense lines with frequencies: 250, 629–632 cm$^{-1}$, corresponding to the A$_1$(TO) fundamental vibrations of the symmetry type along the polar $z$ axis; lines with frequencies 864–872 cm$^{-1}$, corresponding to fundamental vibrations A$_1$(LO) of the symmetry type, and six lines with frequencies of 152, 262, 321–327, 359–362, 429–432, and 578–581 cm$^{-1}$, corresponding to fundamental doubly degenerate vibrations E(TO)-symmetries perpendicular to the polar $z$ axis. It is important to note that in the spectra of the LiNbO$_3$:Gd$^{3+}$(0.003):Mg$^{2+}$(0.65 wt.%) crystal in the range 1000–2000 cm$^{-1}$, upon excitation by a laser line in the visible range (532 nm), we found three broad bands in all the studied scattering geometries. Moreover, one of them has a close spectral position for different geometries.

It is important to note that in the Raman spectra of the LiNbO$_3$:Gd$^{3+}$(0.003):Mg$^{2+}$(0.65 wt.%) in the range of 900–2000 cm$^{-1}$, upon excitation by a laser line with a wavelength of 532 nm, we observed a number of lines of very low intensity in the involved scattering

geometries (Figure 2). At the same time, when the spectrum is excited by the 785 nm laser line, the intensities of the lines in the range of 900–2000 cm$^{-1}$ are higher by an order of magnitude or more (Figure 3). Sufficiently intense lines in the range of 900–2000 cm$^{-1}$, corresponding to the second-order Raman spectrum, were also observed in LiNbO$_3$:Tb(2.24 wt.%), LiTaO$_3$:Cr(0.2): Nd(0.45 wt.%), and LiNb$_x$Ta$_{1-x}$O$_3$ [43–45], characterized by a disordered cationic sublattice and distorted (compared to clusters of a congruent crystal) MeO$_6$ clusters, in the region above 1000 cm$^{-1}$. It should be noted that there are no lines in the recorded Raman spectra whose frequencies are greater than the precise value of the doubled frequency of the 4A$_1$(z)LO mode (872 × 2 = 1744 cm$^{-1}$).

For the LiNbO$_3$:Gd$^{3+}$(0.003):Mg$^{2+}$(0.65 wt.%) double doped crystal, we did not observe a low-intensity line with a frequency of 120 cm$^{-1}$ corresponding to two-particle states of acoustic phonons with a total wave vector close to zero, which is confidently observed in the scattering geometries involved $x(zz, zy)\overline{x}$ and $y(zz, zx)\overline{y}$ in the spectra of all nonstoichiometric LiNbO$_3$ crystals [2,27,29]. The absence of a 120 cm$^{-1}$ Raman line indicates that the R = [Li]/[Nb] value approaches unity, which is characteristic of a nominally net LiNbO$_3$ crystal of stoichiometric composition [2,27,46]. The appearance of this low-intensity line in the Raman spectrum indicates a strong anharmonic interaction of the lowest frequency fundamental A$_1$(TO)-mode (quasi-soft mode) with the acoustic continuum [46]. In the Raman spectrum of the LiNbO$_3$:Gd$^{3+}$(0.003):Mg$^{2+}$(0.65 wt.%), we also did not find any lines corresponding to pseudoscalar vibrations of the A$_2$ symmetry type with frequencies of 209, 230, and 880 cm$^{-1}$, previously recorded in crystals of LiNbO$_3$ heavily doped with magnesium [30]. The appearance in the Raman spectrum of lines corresponding to pseudoscalar vibrations of the A$_2$ symmetry type is possible due to a decrease in the local point group of the symmetry of the crystal from C$_{3V}$ to C$_3$ due to the distortion of oxygen-octahedral MeO$_6$ clusters by dopant cations [2,47]. The optical representation for the point symmetry group C$_3$ is 9A(z) + 9E(x, y) [2]. Thus, pseudoscalar vibrations of the A$_2$ symmetry type, forbidden in the Raman spectrum for C$_{3V}$ by the selection rules, pass into vibrations of the A$_1$(z) symmetry type permitted by the selection rules for C$_3$. The fact that there are no lines in the Raman spectrum corresponding to A$_2$ vibrations of the symmetry type indicates an insignificant effect of the doping Gd$^{3+}$ and Mg$^{2+}$ cations on the geometry of oxygen-octahedral MeO$_6$ clusters in the LiNbO$_3$:Gd$^{3+}$(0.003):Mg$^{2+}$(0.65 wt.%) crystal.

Figure 3 shows the Raman spectra of the LiNbO$_3$:Gd$^{3+}$(0.003):Mg$^{2+}$(0.65 wt.%) crystal obtained in the same backscattering geometries upon excitation by a laser line with a wavelength of 785 nm. From Figure 3, we can see that the Raman spectrum upon excitation by a laser line with a wavelength of 785 nm contains many more lines than are allowed by the selection rules, taking into account the LO-TO splitting for the space group R3c ($C_{3V}^6$), with two formula units in the unit cell [2]. The recorded spectra contain lines corresponding to the fundamental vibrations of the crystal lattice (<900 cm$^{-1}$), as well as second-order Raman lines (in the range 900–2000 cm$^{-1}$). Namely, bands with maxima near the frequencies 1036–1046, 1122–1131, 1202, 1279, 1291, 1351, 1398, 1411, 1522–1525, 1595, 1709, 1720, 1853–1862, and 1963 cm$^{-1}$ correspond to the second-order Raman spectra, the frequencies of which are significantly higher than the frequencies corresponding to the fundamental modes of the crystal lattice, located in the range of 150–900 cm$^{-1}$. These lines correspond to overtone processes (bound states of optical phonons).

Thus, we have found significant differences in the Raman spectra of the LiNbO$_3$:Gd$^{3+}$(0.003):Mg$^{2+}$(0.65 wt.%) crystal obtained upon excitation by laser radiation at wavelengths of 532 and 785 nm. We managed to register the Raman spectra of the first and second orders of the crystal LiNbO$_3$:Gd$^{3+}$(0.003):Mg$^{2+}$(0.65 wt.%). When excited by a near-IR laser line (785 nm), more lines are observed in the recorded second-order Raman spectrum. Differences in the Raman spectra of the LiNbO$_3$:Gd$^{3+}$(0.003):Mg$^{2+}$(0.65 wt.%) obtained upon excitation by laser lines with wavelengths of 532 and 785 nm can be explained via a different mechanism of interaction, with visible and near-IR radiation of microstructural features and structural defects of the crystal, which determine the phonon–

phonon interaction (bound states of phonons) and the anharmonicity of the vibrations of the crystal lattice.

Methods for calculating the bound states of phonons in crystals were proposed earlier in the theoretical works [48–50]. In these papers, many-particle states of phonons are associated with the anharmonicity of vibrations of the crystal lattice. Using Green's function method, one can calculate the density of two-phonon states $\rho_2(\omega)$ as follows. In the simplest approximation, known as the quasi-Newtonian, the density of single-particle states $\rho_1(\omega)$ can be written in the following form:

$$\rho_1(\omega) = a\sqrt{\omega_0 - \omega} \tag{1}$$

Here $a = -\frac{\omega_0 V \sqrt{2\omega_0}}{2\pi^2 s^3}$, $V$—unit of cell volume, $\omega_0$ is the frequency near the center of the Brillouin zone, and $s$—speed of sound.

The one-particle Green's function is written as:

$$D_1\left(\vec{k}, \omega\right) = \frac{\omega\left(\vec{k}\right)}{2}\left[\frac{1}{\omega - \omega\left(\vec{k}\right) + \frac{1}{2}i\Gamma} - \frac{1}{\omega + \omega\left(\vec{k}\right) - \frac{1}{2}i\Gamma}\right] \tag{2}$$

Then, the two-particle Green's function is written as follows:

$$D_2\left(\vec{k}, \omega\right) = \frac{2F(\omega)}{1 - \frac{1}{2}g_4 F(\omega)} \tag{3}$$

where $g_4$ is the anharmonicity constant and $F(\omega)$ is defined as

$$F(\omega) = \frac{i}{(2\pi)^4}\int d^3\vec{k}\int D_1\left(\vec{k}, \omega - \omega'\right) D_1\left(\vec{k}, \omega\right) d\omega' \tag{4}$$

Integrating (4) once, we obtain:

$$F(\omega) = \frac{1}{4}\omega_0^2 a \int\limits_0^\Delta \frac{\sqrt{\omega'}}{\omega - 2(\omega_0 - \omega') + i\Gamma} d\omega' \tag{5}$$

where $\Delta$ is a small part of the dispersion curve of optical phonons. Finally, we can obtain the final expression for $\rho_2(\omega)$:

$$\rho_2(k, \omega) \approx -\frac{2}{\pi\omega_0^2}\frac{\mathrm{Im}F(\omega)}{\left[1 - \frac{1}{2}g_4\mathrm{Re}F(\omega)\right]^2 + \left[\frac{1}{2}g_4\mathrm{Im}F(\omega)\right]^2} \tag{6}$$

As can be seen from Figure 3, there are two second-order Raman lines (1853 and 1963 cm$^{-1}$) of LiNbO$_3$:Gd$^{3+}$(0.003):Mg$^{2+}$(0.65 wt.%), the frequencies of which are noticeably higher than the exact value overtone frequency of the mode 4A$_1$(z)LO (1752 cm$^{-1}$). Let us apply the methods for calculating the bound states of the optical phonons to describe these lines. Figure 4 illustrates a comparison between the experimentally observed intensity of Raman scattering, and a theoretical calculation.

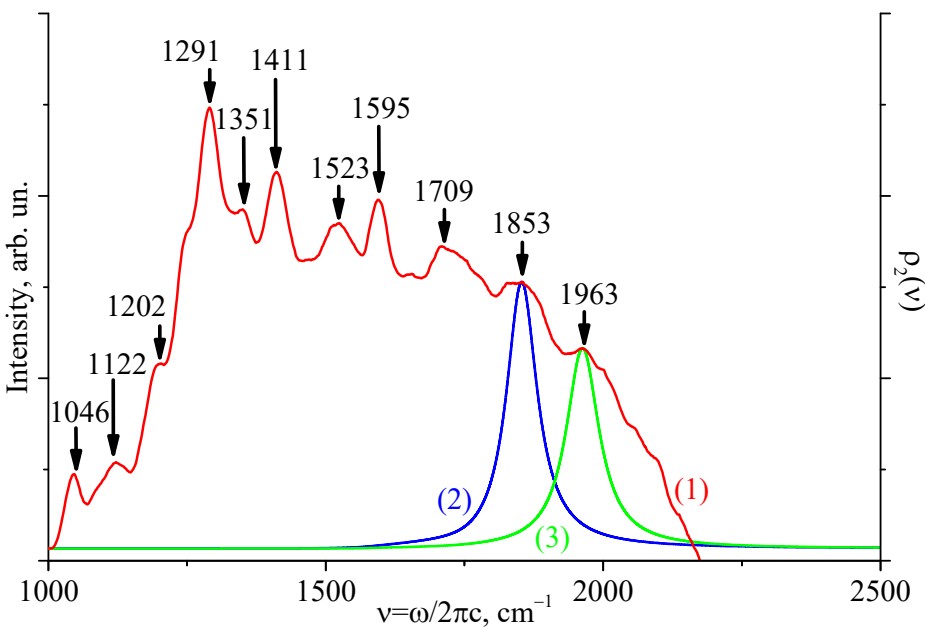

**Figure 4.** Raman scattering spectra of the crystal $LiNbO_3:Gd^{3+}(0.003):Mg^{2+}(0.65$ wt.%) in the region of two-phonon states: 1—experimental curve, 2 and 3—calculated dependences.

As shown in this picture, for the band at 1853 cm$^{-1}$ (curve 2 in Figure 4), a satisfactory agreement for the bound state of the $4A_1$(LO) polar mode with $\nu_0 = 876$ cm$^{-1}$ is achieved at s = 1000, $\Delta = 0.1\ \omega_0$, $g_4 = 4.61 \cdot 10^{-18}$, and $\Gamma = 7.53\ 10^{12}$. For the 1963 cm$^{-1}$ band (curve 3 in Figure 4), a good agreement is obtained at the values s = 1000, $\Delta = 0.1\ \omega_0$, $g_4 = 7.54 \cdot 10^{-18}$, and $\Gamma = 7.53 \cdot 10^{12}$. Near the center of the Brillouin zone, the dispersion curve of the $4A_1$(z)-mode occupies a small area, which explains the choice of the parameter $\Delta$ in Equation (5).

## 4. Conclusions

In this work, the full Raman spectra of the $LiNbO_3:Gd^{3+}(0.003):Mg^{2+}(0.65$ wt.%) crystal, which is a promising material for laser radiation conversion, are recorded in the backscattering geometries $x(zz, zy)\overline{x}$, $y(zz, zx)\overline{y}$, and $z(xx, yy, xy)\overline{z}$. We have given an interpretation of the registered Raman lines, including lines with ambiguous assignments. The Raman spectra show that, in the structure of the $LiNbO_3:Gd^{3+}(0.003):Mg^{2+}$ (0.65 wt.%) crystal, oxygen-octahedral clusters of $MeO_6$ (Me–Li, Nb, Gd, and Mg) are slightly distorted, and in addition, the value of R = [Li]/[Nb] is increased compared to that for a congruent crystal. That is, during doping, an increase in the stoichiometry of the crystal occurs. The fact of the increase in stoichiometry is confirmed by the fact that the Raman spectrum of the $LiNbO_3:Gd^{3+}(0.003):Mg^{2+}(0.65$ wt.%) crystal lacks a low-intensity line with a frequency of 120 cm$^{-1}$, corresponding to two-particle states of acoustic phonons with a total wave vector that is close to zero. Its absence in the spectrum also indicates a small anharmonic interaction of the lowest-frequency fundamental vibration $A_1$(TO)-symmetry type (quasi-soft mode) with the acoustic continuum. A low-intensity line with a frequency of 120 cm$^{-1}$ is confidently observed in the spectra of non-stoichiometric $LiNbO_3$ crystals, nominally pure and doped, and this is absent in the Raman spectra of a stoichiometric crystal [2]. The results obtained indicate a high degree of structural perfection of the crystal and allow us to state that the $LiNbO_3:Gd^{3+}(0.003):Mg^{2+}(0.65$ wt.%) crystal is close in some of its properties to the stoichiometric $LiNbO_3$ crystal. One of the properties of stoichiometric and magnesium-doped $LiNbO_3$ crystals that is important for creating materials for laser radiation conversion on periodically polarized submicron-sized domains with flat boundaries [4–6] is a low value of the coercive field ($\approx$2.3 kV/cm). In a congruent $LiNbO_3$ crystal, the coercive field is much higher, $\approx$23.0 kV/cm.

When the Raman spectrum of a $LiNbO_3:Gd^{3+}(0.003):Mg^{2+}(0.65 \text{ wt.\%})$ crystal is excited by a laser line with a wavelength of 532 nm, the intensities of the lines corresponding to the second-order spectrum are significantly (by an order of magnitude or more) less than the intensities of the second-order lines observed at excitation with a 785 nm laser line. At the same time, in the Raman spectrum in the region below 900 $cm^{-1}$, no lines were found to correspond to the second-order spectrum, as well as lines with frequencies of 209, 230, 298, and 880 $cm^{-1}$ crystals of $LiNbO_3:Mg$ [30]. A comparison of the second-order Raman spectra shows that the near-IR laser line makes it possible to detect more second-order Raman lines.

**Author Contributions:** Conceptualization, N.S. and A.S.; methodology, N.S., M.P. and A.S.; software, A.S.; formal analysis, A.S.; investigation, A.P.; resources, N.S. and M.P.; data curation, A.P.; writing—original draft preparation, A.S., A.P. and N.S.; writing—review and editing, M.P.; visualization, A.P.; supervision, N.S. All authors have read and agreed to the published version of the manuscript.

**Funding:** This study was supported by Ministry of Science and Higher Education Russian Federation scientific topic № 0186-2022-0002 (FMEZ-2022-0016) and the RFBR and BRFBR (grant No. 20-52-04001 Bel_mol_a).

**Institutional Review Board Statement:** Not applicable.

**Informed Consent Statement:** Not applicable.

**Data Availability Statement:** Data on this research will be available from the corresponding author, A.P., upon reasonable request.

**Conflicts of Interest:** The authors declare no conflict of interest.

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
