# Peer review of "Investigation of the Structural Perfection of a LiNbO3:Gd3+(0.003):Mg2+(0.65 wt.%) Double-Doped Single Crystal Using the Raman Spectra Excited by Laser Lines in the Visible (532 nm) and Near-IR (785 nm) Regions"

_applsci, doi:10.3390/app13042348_

Round 1

Reviewer 1 Report

Dear authors,

greetings!

The manuscript “Investigation of the structural perfection of a LiNbO3:Gd3+(0.003):Mg2+(0.65 wt.%) double-doped single crystal using the Raman spectra excited by laser lines in the visible (532 nm) and near-IR (785 nm) regions” focuses on the use of Raman spectroscopy to study features of the structure of a LiNbO3:Gd3+(0.003):Mg2+(0.65 wt.%) crystal.

The experiments were well-designed and performed and the results were presented and discussed in a satisfactory manner. However, there are two aspects that still deserve attention and can be improved.

"Introduction" lacks a detailed description of lithium niobate’s interesting/useful properties such as ferroelectric, piezoelectric, acousto-optical, non-linear optical, and electrooptical ones as same as the chemical, thermal and mechanical stabilities presented by the material. Please, present these features to the readers to justify the importance of studying the material.

Regarding the second aspect, the experimental part related to the material’s synthesis needs to be described in a more detailed way (regarding parameters of the experiment, pieces of equipment used and manufacturer), to allow readers to repeat the process if they want to. Please, dedicate attention to this.

Author Response

Thank you very much for your valuable comments. We tried to take them into account in the new version of the article. Corrections are marked in yellow.

  1. The introduction has been changed in the revised version of the article. Necessary links added.
  2. The section "Materials and Methods" was significantly expanded: details of crystal growth, information about equipment manufacturers, and some essential details of the experiment were added. Non-essential details, such as the internals of both spectrometers, have been removed from the revised version of the article.

Reviewer 2 Report

The paper contribution to the field is not significant as it studies 1 sample and present several spectra measured for this sample. The spectra are not treated in details, the fitting procedure is questionable, figures of poor quality. Authors claim that they study LiNbO3 doped Mg and Gd, but the sample contains many other impurities with similar or higher concentrations ! The crystals were  grown from nonstoichiometric melt and their concentration was overinterpreted as being stoichiometric (I can see from given spectra in the paper that crystals are not stoichiometric).The mode identification of LiNbO3 is not correct. The reference works in the field are not cited. The second order can be seen in the Raman spectra excited with visible laser ! There are many auto citations and citations of close collaborators. The introduction part of the paper do not express the motivation for such study. 

Author Response

Thank you very much for your valuable comments. We tried to take them into account in the new version of the article. Corrections are marked in green.

  1. The assignment of the first-order Raman lines of the studied crystal has been corrected in accordance with articles 36 and 37 from the list of references.
  2. A detailed description of the conditions for obtaining a LiNbO3:Mg:Gd crystal has been introduced in the “Materials and Methods” section. It should be noted, in response to the reviewer's remark, that in any doped LiNbO3 crystals, the [Li]/[Nb] ratio will never be equal to unity, even if dopants are introduced into the melt with a strictly stoichiometric composition [Li]/[Nb]=1. In our case, the congruent crystal was directly doped ([Li]/[Nb]=0.946). When a LiNbO3 crystal is doped, doping metals (Me) fall into vacant octahedra and also displace lithium and niobium from their positions, forming point defects MeLi, MeNb, and MeV, significantly violating the stoichiometry. In this case, there can be both an increase and a decrease in stoichiometry. For example, at high concentrations of the alloying metal magnesium (>3 wt.%) in a congruent melt, niobium is predominantly displaced by magnesium from the oxygen octahedra, which leads to an increase in the stoichiometry ([Li]/[Nb] values) of the crystal. In double-doped crystals, the situation is even more complicated and far from unambiguous. On this occasion, the corrected article for the investigated crystal in the Introduction section is given an explanation.
  3. In the revised version of the article, we tried to reduce the number of self-citations. Unfortunately, some references to our works cannot be removed, since they describe the results obtained only by us. Similar studies were not carried out in the works of other authors. In addition, due to the desire of the reviewers to strengthen Section 2 (“Materials and Methods”), we are forced to give 3 references to our works, where a detailed description of the conditions for obtaining the charge and LiNbO3:Mg:Gd single crystals is given.
  4. You are right: when the Raman is excited by green radiation, second-order lines are visible. Corresponding corrections have been added to the text.
  5. In the revised version of the article, the "Introduction" section has been expanded to strengthen the motivation for this study.

Reviewer 3 Report

In this work, the authors performed the Raman spectroscopic study on co-doped LiNbO3:Ga, Mg single crystal excited by the visible and NIR lasers. They found that the 2nd-order Raman scattering can be excited by the 785-line laser source. And these active Raman modes are polarized that can only be detected by the certain scattering geometries. The co-doped LN:Ga, Mg crystal possessed the slight distortion and was close to the pure stoichiometric crystal. There are some comments should be addressed before acceptance as following:

1. In the section of introduction, the authors claimed that “the more perfect the crystal structure, the less intense the 2nd-order Raman spectrum should be”. This is conflict with the common sense, because the 1st-order Raman intensities were in positive relationship with the crystalline quality and the 2nd-order Raman intensities were dependent upon the 1st-order ones. Could the authors explain the reason as described in the manuscript?

2. A schematic diameter of the unit cell of crystal as well as the vibration modes in the cell is recommended to provide in the manuscript, in order to help the audience understanding the configurations and discussion.

3. The close-up views of the spectra in 1000-2000 cm-1, similar to Figure 2a, should also be provided in the other subfigures in Figures 2 and 3, to highlight the 2nd-order Raman scattering only observed by the NIR excitation and certain geometry.

4. In table 2, why the wavenumbers of Raman modes with the same geometry are different under 532 and 785 nm excitation. It should be independent from the excitation wavelength.

5. In Figure 3, only two Raman peaks were numerically predicted and compared with experimental results. How about the other 2nd-order Raman peaks?

6. The reason of the polarized 2nd-order Raman scattering and certain scattering geometry for activation should be interpreted in the manuscript. It is of importance to understand the physical properties of the co-doped LN crystal.

Author Response

Thank you very much for your valuable comments. We tried to take them into account in the new version of the article. Corrections are marked in cyan.

  1. In the revised version of the article, we have eliminated this contradiction.
  2. A schematic of the unit cell of lithium niobate as well as images of the fundamental vibrations are presented in the literature, for example, in the Ref. 36.
  3. We carried out additional experiments and a thorough analysis of the spectra. It turned out that, when the RS is excited by green radiation, second-order lines are visible. Corresponding corrections have been added to the new version of the article. Zoomed-in inserts are also added to Figures 2 and 3 to show the second-order Raman spectra.
  4. Only in the ideal case, and in particular for an ideal defect-free non-photorefractive crystal, the main parameters of the lines in the Raman spectrum (frequency, line width, and intensity) do not depend on the wavelength of the exciting laser radiation. In real crystals, due to the presence of effects of strong disordering of the crystal lattice (intrinsic and photoinduced), the situation is much more complicated. As is known, lithium niobate is not only a deeply defective oxygen-octahedral phase of variable composition, but also a photorefractive crystal. In this regard, when a crystal is illuminated with laser radiation, an additional volumetric ordered sublattice of nano- and microstructures with fluctuating refractive index, dielectric permittivity, conductivity, and other parameters that differ from the corresponding parameters of a single crystal in the absence of the photorefraction effect appears in the illuminated region of the crystal. That is, the effect of photorefraction makes an additional contribution to the disordering of the crystal structure. As a result, the Raman scattering lines are shifted in frequency, their intensity changes, and they broaden. In addition, in a lithium niobate crystal, as a phase of variable composition, and in the absence of the photorefraction effect, according to X-ray diffraction analysis, there is a superstructure - an ordered sublattice of intrinsic defects. In addition, when polarized laser radiation passes through a photorefractive crystal, its depolarization also occurs, which depends significantly on the magnitude of the photorefraction effect. As a result, if the Raman spectrum and photorefraction are excited by the same laser radiation, then the appearance in the sample under study due to photorefraction of radiation that differs from the laser radiation incident on the crystal, both in the direction of propagation and in the direction of polarization, should lead to the appearance of lines in the Raman spectrum ( corresponding to both fundamental vibrations and the second-order spectrum) forbidden by the selection rules for a given scattering geometry. In this case, the intensity of the "forbidden" lines in the Raman spectrum of a photorefractive crystal will increase with an increase in the photorefraction effect. As is well known, the magnitude of the photorefraction effect depends on the power density of the laser radiation and on the wavelength of the laser radiation. As the wavelength of the exciting laser radiation increases, the magnitude of the photorefraction effect decreases. When Raman spectra are excited by laser radiation in the near-IR region, there is no photorefraction effect, i.e., there is no effect on the Raman spectrum. For the reasons listed above, we carried out comparative studies of the Raman spectra upon excitation by radiation in the visible range (532 nm, the photorefraction effect is maximum) and upon excitation by a 785 nm laser line (there is no photorefraction effect). Thus, the Raman spectra of a photorefractive lithium niobate crystal when excited by laser radiation at 532 nm and 785 nm should differ in frequency, intensity, line width, and number of lines in the spectrum. Such differences were found in our work. It should also be noted that non-photorefractive additives cannot completely suppress the effect of photorefraction in a lithium niobate crystal.
  5. We paid great attention to the second-order Raman lines with the highest frequencies (1853 and 1963 cm–1) due to the fact that their frequency exceeds the exact value of the doubled frequency of the overtone of the 4A1(LO) line (2×876=1752 cm–1) .
  6. In the new version of the article, we have updated the experimental Raman spectra and the corresponding discussion.

Round 2

Reviewer 3 Report

I have no more questions on this manucript.

Author Response

Thank you very much for your useful comments.